# Is social cohesion produced by weak ties or by multiplex ties? Rival hypotheses regarding leader networks in urban community settings

**Silvio Salej Higgins** [1] *, **Neylson Crepalde** [2], **Ivan L. Fernandes** [3]

**1** Associated Professor of Sociology, Federal University of Minas Gerais, Belo Horizonte, Brazil, **2** Professor of Data Science, Pontifical Catholic University of Minas Gerais, Belo Horizonte, Brazil, **3** Research Assistant, Federal University of Minas Gerais, Belo Horizonte, Brazil

* sisahi@yahoo.com

## Abstract

In his seminal work, Mark Granovetter (1973) challenged sociologists to test sociometric hypotheses regarding collective action in communitarian settings. In this article, we tested the two main hypotheses which consider social cohesion in communitarian urban settings– these being firstly cohesion by weak ties and secondly cohesion by multiplex ties. We studied the elite leaders of two slum communities of Belo Horizonte (Brazil). Three social processes were examined as multiplex interactions: recognized status, exchange of useful information and collaboration. Our findings reveal, on the one hand, that multiplexity is associated with the frequency of ties and, on the other, that reciprocity and shared domains of performance fuel such strong multiplexity. If we assume that elite connections conform to a high order structure, our findings, in contrast to previously well-established hypotheses, reveal a segmented social order in which multiplexity does not mean the overlapping of social circles. On the contrary, multiplexed social exchanges are restricted to specialized domains.

## Introduction

At the end of his seminal article, Mark Granovetter [1] encouraged the sociological community to test new hypotheses concerning how weak ties work in order to improve collective action. When considering the urban public policies launched in Boston in the 1950s, he proposed a set of new observations on social cohesion of urban segregated communities. These insights revealed (a) a community without rich networks of weak ties would not be capable of collective action; (b) if organizational life and workplace are the sources of weak ties, then bedroom neighborhoods would not be able to create bridge ties beyond the immediate inner circles of people; (c) the lack of weak ties affects confidence between leaderships and the grassroots.

In a technical sense, we must remember that Granovetter [1] defines the strength ties as simultaneously a function of frequency, intimacy of interaction and as structural property of weak ties. This is a consequence of his axiom: the forbidden triad. That is, a triad composed of

**Funding:** Higgins S.S Grant APQ-02260-14 Fundação de Amparo à Pesquisa de Minas Gerais (Brazil) FAPEMIG https://fapemig.br The funders had no role in study design, data collection and analysis, decision to publish, or preparation of the manuscript.

**Competing interests:** The authors have declared that no competing interests exist.

strong ties (high frequency and intimacy) will be a closed clique where the shortest distance is $d = 1$. Hence, any alternative path for overcoming the isolation of strong ties will be at least $d = 2$ or more. In this vein, we can say that the structural property of weak ties is not separable from the inner nature of edges.

From Granovetter's perspective [1], the aforementioned conjectures would be useful for understanding the critical case of the West End—an Italian community situated in Boston: The question is why was this community unable to address a public top-down project that dramatically affected local life?

This question, grounded in the former set of conjectures, merited a reply *in extenso* from Herbert Gans [2]. The *American Journal of Sociology* understood the importance of the problem and published this exciting sociological debate, which was somewhat unusual in the sociological setting [2–4]. From Gans' point of view [2], only middle-class communities would be able to create a cohesive confidence between leaders and the grassroots. Granovetter [3] questioned why this was. Despite his recognition of heuristic power in the weak ties hypothesis, Gans [4] highlights some of Granovetter's misunderstandings of the social context of the West End Community [3]. Firstly, there were weak ties, but the community was fragmented in space. Secondly, weak ties depend on historical and cultural factors. In the West End, there was no tradition of social struggles seeking to improve the welfare of the community. For example, anyone who opposed the project would be disapproved of by their peers because the Catholic Church supported the Government's urbanistic intervention in local life. Finally, the only leader working in the neighborhood was a white man who was mistrusted by black people. As a balanced conclusion, Granovetter [3] accepted that weak ties were both the cause and consequence of history and culture.

Granovetter [1] was convinced of the theoretical power of his framework in terms of prediction of collective action, taking into account the extent to which weak ties bridge clustered communities, albeit recognizing the methodological limits of his challenge:

"In the absence of actual network data, all this is speculation. The hard information needed to show either that the West End was fragmented or that communities which organized successfully were not, and that both patterns were due to the strategic role of weak ties, is not at hand and would not have been simple to collect. Nor has comparable information been collected in any context" (Granovetter, [1])

## Later studies on community networks

After its original formulation, the hypothesis of weak links has been thoroughly examined by social scientists. Susan Greenbaum [5] developed a state-of-the-art treatment on the mechanisms of cohesion in urban communities. Going beyond the debate between Gans and Granovetter [2–4], she sought evidence to test the cohesive strength of weak bonds, by utilizing Wellman's [6] findings on the role of intimate relationships in sparse networks beyond the local community. Based on a survey performed in four urban communities of worker strata in Kansas, Greenbaum proposes an alternative hypothesis to that formulated by Granovetter [1]: bridge ties, vital for the general cohesion of an urban community, consist of multiplex type interactions, in which family relationships can be superimposed onto co-participation in community organizations, as well as being endowed with affective intensity. In other words, strong multiplex-type ties can be bridge ties.

The input from these authors indicated that the communities studied were made up of interaction networks linked to spatial proximity. In the first place, there were intra block-face networks formed of neighbors of residential units living close to each other. In this type of cluster, the main ties between acquaintances were latent relationships that fit the concept of

weak ties proposed by Granovetter [1]. Secondly, there were inter block-face networks, composed of multiplexed loops where relationships of kinship, belonging to the same company, club or church, old childhood gangs, etc. were superimposed. In summary, strong-multiplex ties, between spatially sparse neighborhoods, can be understood as a source of greater cohesion between social clusters, facilitating effective communication on a broad geographical scale and a greater sense of identity beyond face-blocks.

In summary, Greenbaum [5] found two pieces of evidence which cast doubt on Granovetter's hypothesis [1]:

- Weak ties structure clusters spatially close.

- Strong multiplex ties bridge spatially dispersed clusters.

Many years later, Robert Sampson [7, 8] developed a pioneering study that corresponded to the two elements of Granovetter's puzzle [1]: the theoretical role of weak ties in bridging clustered communities and the methodological strategy for collecting network data. In *Leadership and the higher-order structure of elite connections*, Sampson [8] proposed a strategy for understanding the covariance between the collective efficacy of communities and the network structure of elite leaders in the city of Chicago. In order to understand the endogenous capability for collective action in a community, Sampson constructed an index entitled *Collective Efficacy*, integrating attitudinal and objective factors. Survey data is the informational basis of the index. For example, basic survey questions aim to discover whether neighbors react to observed misconduct such as drug use or antisocial acts against public amenities.

In Sampson's research strategy [8], there are several elements that need to be highlighted in terms of how he addresses Granovetter's puzzle [1]. Firstly, the focus was on what Sampson called "high order structure" [8]. This corresponds to an interactional structure among selected people with at least two degrees of distance, that is, supra dyadic connections. In the empirical field of Chicago City, what was considered high order was the elite leaders identified by a mixed strategy that included secondary data and snowball sampling. In this sense, there was a double problem to be solved. The first of these was how to identify those who were considered people who "get things done" in the community. The second was to identify what the boundary of that social universe called "elite" which would be targeted by the researcher. Sampson's insight stated that a network of key leaders creates systematic and influential connections both within and among communities.

Secondly, the sampling plan identified six different realms in which a citizen could be considered a leader: education, politics, religion, business, law enforcement, and community organization. Forty-seven community zones were selected from within all of the social strata of the 77 administrative zones into which the city was divided.

Thirdly, a *geocoded list* was created with 10,000 names of individuals identified as leaders. The data was collected from diverse sources such as telephone books and also business and service directories. Fourthly, around 5,500 leaders were identified in the 47 areas, and attributes of personal identity, work and location were collected. Fifthly, 2,500 cases were selected and stratified by community and realms. Sixthly, more than 1,700 interviews were conducted from among these selected cases. Lastly, snowball sampling was performed in order to generate new names, asking who the most influential people in the six realms were. In the end, more than 3,800 new names were generated.

The former sampling process was repeated as a panel survey seven years later, in 2002. This second round aimed to investigate: (a) the permanence of leaders; (b) the emergence of new leaders in old positions or in new organizations; (c) the trajectory of leaders. Due to a decreased budget, only 30 communities out of the former 77 were included in the new sample.

A high turnover was found because only 60 percent of respondents were in positions similar to those of the first round.

This overview of Sampson's strategy [8] enables us to better understand the two challenges posed by Granovetter [1]. When he said that there was no available data to test the weak ties hypothesis, he was referring to a major problem in social network analysis: How can a structured set of interactions be sampled? This is not a trivial problem because structural research runs opposite to standard survey sampling. Intrinsically, a network is a structure. Thus, the researcher must assume that its elements are interdependent. In contrast, standard survey sampling works on the assumption of independent observations. The latter is the elementary heritage of the positivistic point of view that constrains subjective bias when the researcher is observing the social world [9, 10].

At this point, some limitations in Sampson's strategy must be highlighted. The elite network is not a representative sample in the standard meaning of survey research. The universe of leaders and their interactions is a specific sample chosen by the observer. As with any social network research, when defining the boundaries of the object, the researcher considers some plausible criteria such as the seven steps mentioned earlier. Consequently, we know nothing about the structural relations between the elite and grassroots, which was one of the key puzzling problems presented by Granovetter [1]. Nevertheless, given the state of the art of network sampling techniques, we could not conceal this tradeoff between boundary and representation [11].

In theoretical terms, it is important to note that the debate on social cohesion, reviewed here, leaves aside the spatial approach to the problem of community cohesion. Recent research, from a geographical perspective, has emphasized the difference between spatial and relational proximity. In this way, the concept of community is expanded beyond the place of living or working and is extended to the sense of organizational belonging [12–14]. However, the present research does not focus on this discussion as it did not collect data on the location of the leadership in the physical space of the city.

## Rival hypotheses

Broadly speaking, the seminal hypothesis proposed by Granovetter [1], affirms that weak ties operate as bridging and cohesive links in spatially sparse networks, which we refer to as *global cohesion by weak ties*. On the one hand, the rival and narrow hypothesis, as stated by Greenbaum [5], argues that multiplex ties operate as bridging and cohesive links in spatially sparse networks, which we refer to as *global cohesion by multiplex ties*. On the other hand, weak ties function as bridging and cohesive links in spatially close networks, which we refer to as *local cohesion by weak ties*. Table 1 depicts the set of aforementioned theoretical hypotheses. With these options available, it would be possible to model a set of data under a triple condition: with multiplex data, spatial information, and data about the strength of ties. In the following, we reveal how we were able to address the challenges of these rival hypotheses.

## Our object: Two multiplex systems of social status in impoverished urban communities

Addressing Grannovetter's puzzle [1] and inspired by previous studies, two impoverished urban communities in Belo Horizonte (Brazil) were selected, in order to better understand

**Table 1. Network data and hypothesis on social cohesion.**

| | Strong multiplex ties | Weak ties |
|---|---|---|
| **Global space** | *Global cohesion by multiplex ties* [5] | *Global cohesion by weak ties* [1] |
| **Local space** | | *Local cohesion by weak ties* [5] |

social exchange and interdependencies between the elites of their leaderships. Methodologically, we went further than Greenbaum [5] and Sampson [7, 8]. In contrast to the former, who worked with ego-networks by employing survey techniques, we collected a complete set of multiplex networks. The advantage of working with a well-defined boundary, where each one of the respondents could indicate all their *alteri*, is well known [15]. Furthermore, we took into consideration the multiplex nature of interactions between leaderships. That is, an interaction process, between two or more actors, which is simultaneously a flux of different resources and an interpretive exchange of different meanings. At the beginning of sociological work, Simmel [16] stressed how socialization was a multiple and simultaneous process across social circles, such as family, school, friendship, work, leisure etc. Recently, in a technical sense, White [17–19] has developed tools for understanding the social structure of multiple networks.

In our study, we collected information on three different social processes from among leaderships of two urban communities: perceived status, exchange of useful information, and coordination/collaboration. We applied three sociometric generators (perceived status, exchange of useful information, and coordination/collaboration) whose answers could be modeled with the same statistical tool: Exponential Random Graph Model (ERGM). In contrast to both Greenbaum and Sampson, in the first sociometric generator we collected proxy data concerning the strength of ties using an interaction frequency scale.

Both communities had a similar profile of income, infrastructure, and public services. However, there was a marked difference in terms of their history. The Alpha community, in contrast to the Beta community, had a long history of violence and homicides related to drug trafficking, as reported in the official database of the police. This is useful background information that permits a more accurate better interpretation of the relational dynamic among leaders.

**Data collection.** Given the state of the art network sampling techniques, we had two methodological options: (1) to sample probabilistically the network structures of two urban sectors of Belo Horizonte (Brazil) with a high density of people (Alpha = 17.000 and Beta = 20.000); (2) to track the elite of leaders using a snowball process with key informants. We did our best when following the second option. In reality, the first option would have demanded an extremely expensive sample [20].

In order to identify the elite leaders, snowball sampling was conducted in both communities using two protocols. The first protocol differentiated eight key community realms in which endogenous leaders operated: education, religion, politics, security, health, business, local organization, sports, and leisure. As a first step, a seed of leaders' names was generated by interviewing key informants in hospitals, schools, grocery stores, associations, churches and so on for each realm. Each respondent named five people they perceived as influential in the community. As a second step, a new round of interviews was conducted to generate another five names. The process ended when we were unable to find new indications, that is, by saturation. The second protocol, with the complete roster of leaders in each community, formulated three sociometric questions on different social processes that were relevant in the exchange and local recognition among leaders. The following were the main processes studied with the respective sociometric questions:

• Perceived status among leaders

*Considering the last twelve months. Of the people on this list, who do you consider to be leaders that have worked in a helpful way for the benefit of the community? You can choose up to ten names.*

The limit of up to ten *alteri* aimed to overcome problems with memory of respondents encouraging them to indicate a reasonable number of partners. That constraint was included in the statistical model.

*How often have you spoken with each of them*?

*Weekly*

*Fortnightly*

*Monthly*

*Bi-annually*

*Annually*

This scale of frequency was not applied in each question because some respondents couldn't see the sense in answering the same question twice. It was not easy for them to disentangle frequencies of interactions regarding the exchange of useful information and the collaboration.

• Exchange of useful information

*Considering the last twelve months. Of the people on this list, who have you turned to in order to request any kind of useful information for your work with the community*? *You can choose up to ten names. Why did you choose each of them*?

• Coordination/Collaboration

*Considering the last twelve months. Of the people on this list, who have you contacted to organize any activity for your community (such as improvements to the school, cleaning the square, helping in the vaccination campaign, etc.)? You can choose up to ten names.*

Two sets of data were collected in each community: firstly, a multiplex network—that is, different interactional structures between the same agents; secondly, relevant attributes on nodes and relationships. The following Table 2 summarizes the data collected:

The scale of frequency in the first sociometric generator, when added to multiplexity of networks, enabled us to operationalize a common construct for the strength of ties as can be inferred from Granovetter's seminal work [1]. On the one hand, "(...) the strength of tie is a (probably linear) combination of the amount of time, the emotional intensity, the intimacy (mutual confiding), and reciprocal services which characterize the tie" [1], p. 1361; on the other hand, footnote 3 "Some anthropologists suggest multiplexity, that is, multiple contents in a relationship, as indicating a strong tie. While this may be accurate in some circumstances, ties with only one content or with diffuse content may be strong as well" [1] p. 1361).

Despite the fact that we do not have any information regarding emotional support among leaders, which is a key element in the canonical definition of strength [1], we consider the frequency of interactions as a necessary condition of the strength of ties in the context of community activity. The leaders, identified endogenously by key informants in the neighborhood, shared a social space without a specific division of labor. Mutual and emotional support is not intrinsic to these leaders' voluntary activities. The original scale, from one to five, was included in the model as following: annually (1), bi-annually (2), monthly (3), fortnightly (4), and weekly (5).

Once again, we must remember that, in this research, we did not collect data on how strong or weak the relations between the perceived leaders and grassroots are. A survey sampling

**Table 2. Data collected.**

| Multiplex Networks | Attributes on nodes and relationships |
|---|---|
| Status | Realms of leadership |
| Information | Frequency of ties |
| Coordination/Collaboration | |

**Table 3. Cohesive metrics [22].**

|  | Density | | Avg Distance | | Compactness index | | Diameter | |
|---|---|---|---|---|---|---|---|---|
|  | **Alpha** | **Beta** | **Alpha** | **Beta** | **Alpha** | **Beta** | **Alpha** | **Beta** |
| **Status** | 0.657* | 0.708* | 1.924 | 2.124 | 0.508 | 0.494 | 4 | 5 |
| **Information exchange** | 0.104 | 0.114 | 2.666 | 2.424 | 0.328 | 0.302 | 7 | 6 |
| **Collaboration** | 0.089 | 0.143 | 3.004 | 2.567 | 0.309 | 0.399 | 7 | 7 |

*Average degree of valued edges.

strategy aimed at tracking the interactions between the elite and the neighborhood would pose significant and almost intractable problems with missing data [21]. We identified one perceived elite of leaders and then relevant data regarding exchanges among them was collected. With previous warnings in mind, and taking into account Granovetter's seminal puzzle [1], we can test the rival hypotheses about social cohesion.

## Descriptive results

Some descriptive statistics enable the comparison of both groups of elite leaders with regards to the level of cohesion in each of the networks. The Alpha Community has an elite of 32 leaders and the Beta Community has 40. Only three social processes—perceived status, search for useful information and collaboration—were included in this analysis, due to issues of accuracy and completeness of data for both communities. We prepared three different square matrices with the respective number of lines and columns.

Table 3 depicts, in a comparative way, how cohesive the three different processes studied are. In both communities, the status system reveals a similar intensity of recognition exchange. In terms of information exchange and collaboration, the Alpha community presents a slightly weaker density if compared with the Beta community. The intrinsic characteristics of each social process could explain this difference. For example, being recognized or recognizing others as leaders does not depend on actual collaborative relationships.

When comparing the frequencies of interactions (Figs 1 and 2), in the Alpha Community 60 percent of the ties show a monthly or higher frequency, whereas in the Beta Community

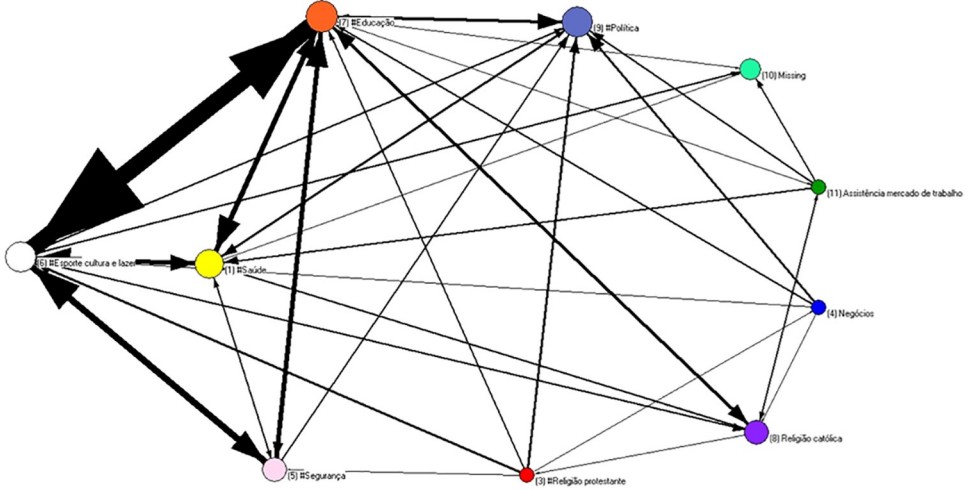

**Fig 1. Alpha community strength of ties by frequency.**

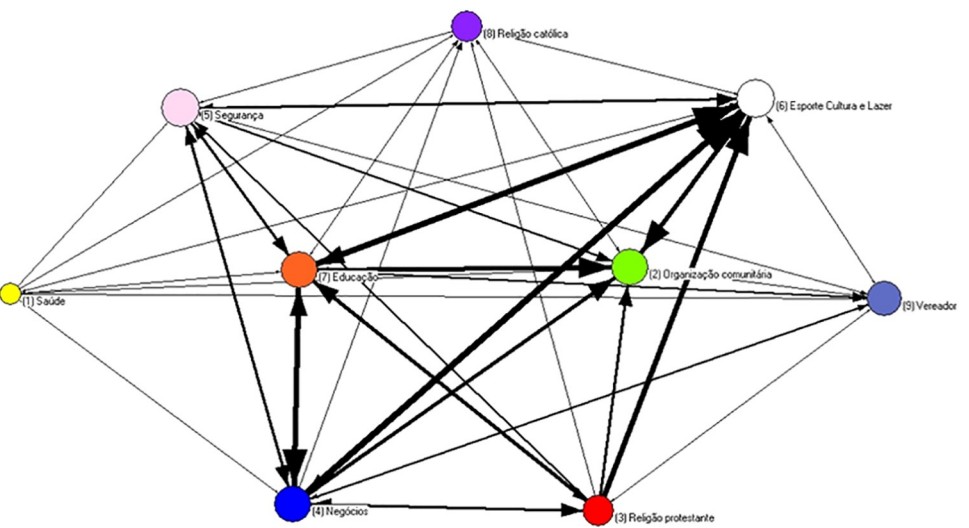

**Fig 2. Beta community strength of ties by frequency.**

this performance is 76.11 percent–scale: 1 = annually, 2 = bi-annually, 3 = monthly, 4 = fort-nightly and 5 = weekly. This could suggest a closer status interactive system in the Beta Community.

In order to explore the exchanges between status realms, the actors' nodes were collapsed. In the Alpha Community (Fig 3), there is an intensive social circuit among leaders operating in the areas of education, safety, health, sports, culture, and leisure. This digraph is a useful tool for identifying what realms are crucial for community life. In the Beta Community (Fig 4), the main social circuit includes some realms that are not important in the previous one, such as local organization, business, or protestant religion. In this case, we distinguished one Catholic priest from Pentecostal pastors. Health and safety are peripheral in this case.

In terms of multiplexity, the two digraphs represent interactions among leaders encompassing the three social processes: status, exchange of useful information and collaboration. An option for

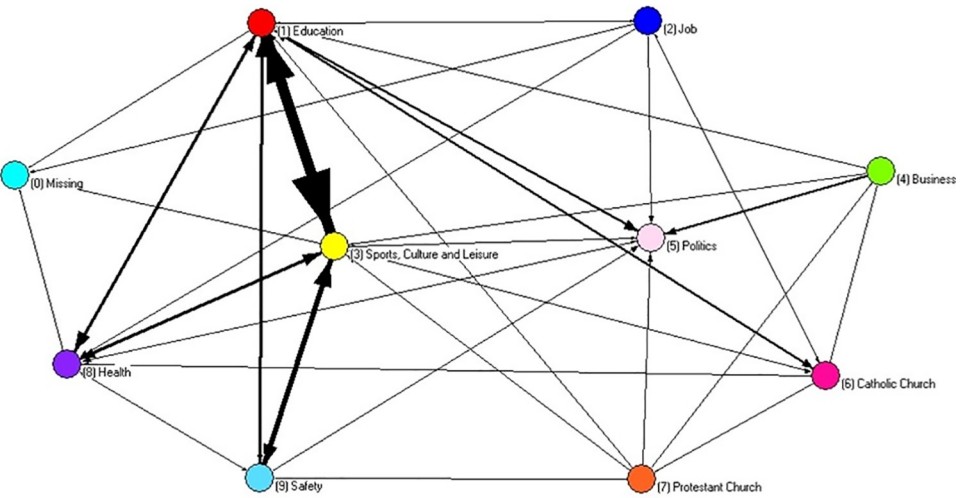

**Fig 3. Digraph of alpha community nodes have been collapsed by realms.**

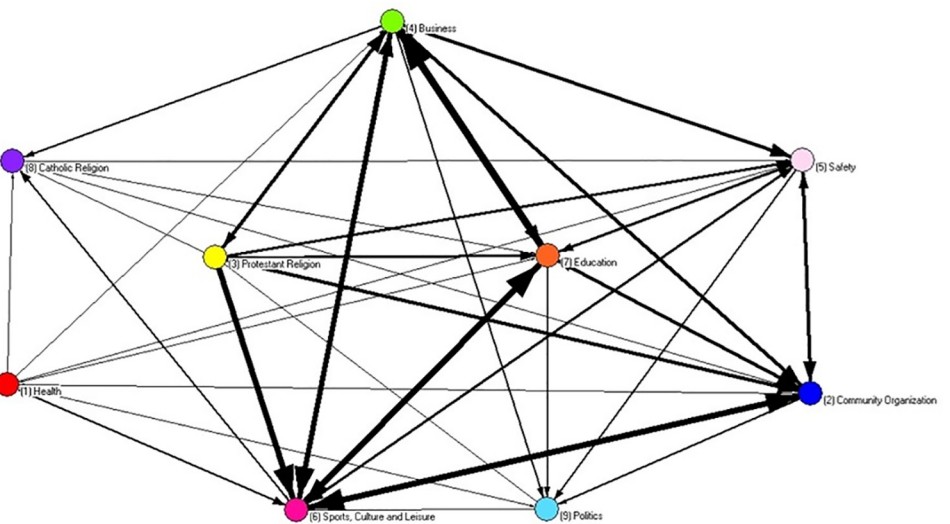

**Fig 4. Digraph of beta community nodes have been collapsed by realms.**

the strict sense of multiplex ties was included. As can be seen, some leaders became isolated after having been cut out of the single and double interactions. In a substantive way, the resultant digraphs can be interpreted as the cohesive core of both social systems (Figs 5 and 6).

## Exponential Random Graph Models (ERGM) to test hypotheses on cohesion

Taking into account that we only have data on two out of the four conditions supposed by the rivalry between Granovetter and Greenbaum [1–5], we posed the following question, in the Table 4, that permits the contrast of our evidence with the previous results of sociological literature.

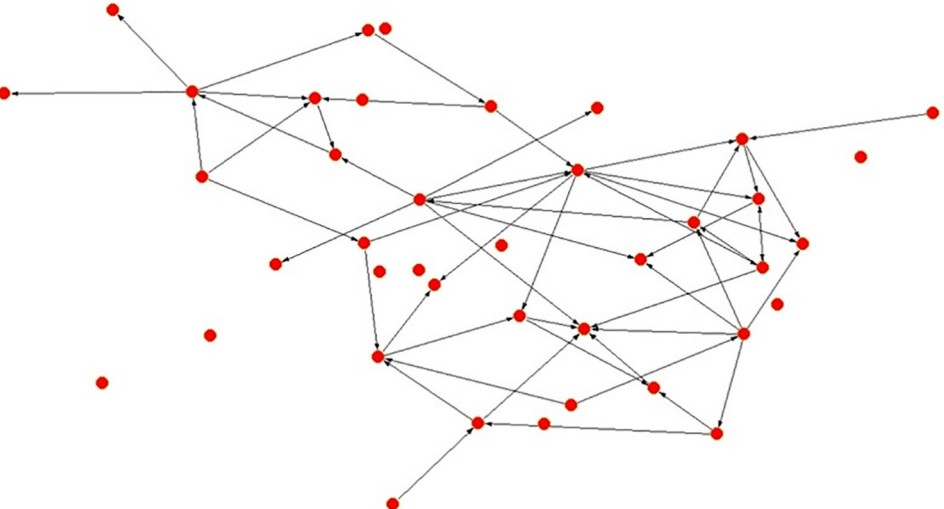

**Fig 5. Multiplex digraph of alpha community an edge represents three interactions: Status recognized, exchange of useful information and collaboration.**

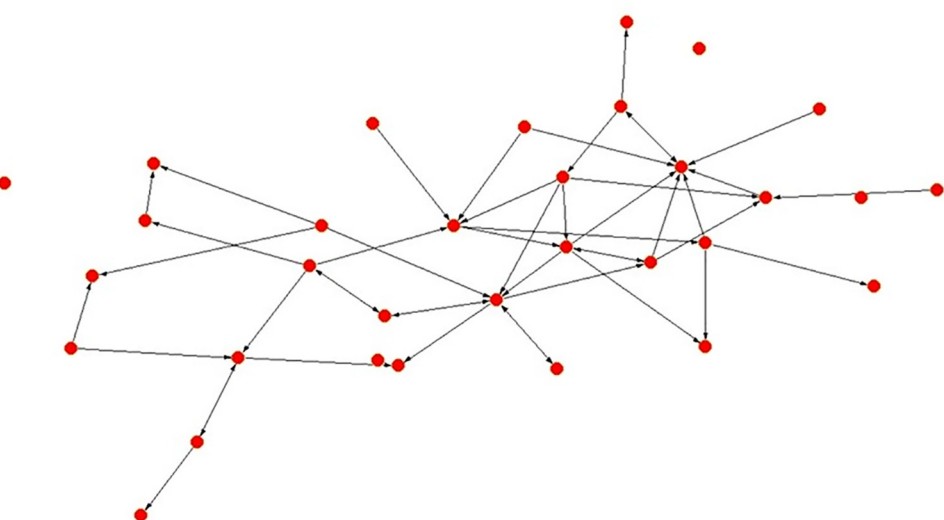

**Fig 6. Multiplex digraph of beta community an edge represents three interactions: Status recognized, exchange of useful information and collaboration.**

**Table 4. New question on elite cohesion.**

| | Frequency of ties |
|---|---|
| **Multiplexity** | *How do the strength of ties (multiplexity and frequency) and other endogenous/exogenous factors determine the emergence of higher-order structure of elite connections?* |

**Table 5. Alpha and beta communities Pearson correlation between multiplexity and strength of ties [22].**

| | Alpha | Beta |
|---|---|---|
| | Multiplex network | Multiplex network |
| Dyadic covariate frequency of ties | 0.452* | 0.438* |

*P value < 0.002.

In an attempt to answer the former question, we modeled the higher-order structure of elite connections. The idea was to understand what factors, whether endogenous or exogenous, determine the emergence of a high order structure among social leaders. Thus, we collected the data posed as a cross-sectional study which was not present in either Granovetter's or Greenbaum's work [1–5]. We asked if the network of leaders is a combinatory result of the strength of ties, measured by multiplexity and frequency, realms of action and some patterns of interaction.

As a preliminary model, we tested the linear correlation between the multiplex ties and the dyadic covariate strength of ties. In Table 5, the Quadratic Assignment Procedure indicated a clear correlation of variables for both communities:

The Pearson coefficients of 0.452 and 0.438 can be considered as positive high correlations between multiplex ties and weekly interaction. In other words, multiplexity and frequency are associated as indicators of the strength of ties.

**Table 6. Multiplexity in the alpha community [24].**

|  | Estimate | Std. Error MCMC | Z value | Pr(>Z) |
|---|---|---|---|---|
| multiplex edges | -4.09958 | 0.63907 | -6.415 | < 1e-04 *** |
| edgecov.weight | 0.78201 | 0.09173 | 8.525 | < 1e-04 *** |
| isolates | -0.4803 | 0.77618 | -0.619 | 0.536044 |
| mutual | 2.45539 | 0.6883 | 3.567 | 0.000361 *** |
| mutual.cat | -2.90803 | 1.30126 | -2.235 | 0.025431 * |
| nodematch.cat | 1.13404 | 0.43908 | 2.583 | 0.009802 ** |
| gwesp.fixed.0.693 | -0.37368 | 0.35699 | -1.047 | 0.295208 |
| gwideg.fixed.0.693 | -0.82859 | 0.69373 | -1.194 | 0.232318 |
| gwodeg.fixed.0.693 | -0.88875 | 0.75741 | -1.173 | 0.240637 |

Signif. Codes:

'***' 0.001

'**' 0.01

'*' 0.05.

The general form of the ERGM model is the following:

$$\Pr(X = x|\theta) \equiv P_\theta(x) = \frac{1}{k(\theta)} \exp\left\{\theta_1 z_1(x) + \theta_2 z_2(x) + \cdots \theta_p z_p(x)\right\} \quad (1)$$

The probability of each parameter is:

$$\Pr(X_{ij} = 1|\theta) = \frac{\exp\theta}{1 + \exp\theta} \quad (2)$$

## Results

ERGMs are useful for understanding the endogenous social process that reveals an interactional social system [19–23]. Tables 6 and 7 depict the high order structure of elite connections for both communities, followed by the respective Goodness of Fit for both models (Figs 7–16). Each model includes two kinds of effects: endogenous–multiplex edges, isolates, mutual, edge covariate with frequency—and exogenous–categorical attributes with the realms of interaction.

**Table 7. Multiplexity in the beta community [24].**

|  | Estimate | Std. Error MCMC | Z value | Pr(>Z) |
|---|---|---|---|---|
| multiplex edges | -5.04055 | 0.65509 | -7.694 | <1e-04 *** |
| edgecov.weight | 0.93271 | 0.09301 | 10.028 | <1e-04 *** |
| isolates | 1.75495 | 0.759 | 2.312 | 0.0208 * |
| mutual | -0.09238 | 0.77727 | -0.119 | 0.9054 |
| nodematch.cat | 0.86339 | 0.38985 | 2.215 | 0.0268 * |
| gwesp.fixed.0.693 | -0.29329 | 0.39505 | -0.742 | 0.4578 |
| gwideg.fixed.0.693 | 0.69849 | 0.76559 | 0.912 | 0.3616 |
| gwodeg.fixed.0.693 | -0.99975 | 0.63851 | -1.566 | 0.1174 |

Signif. Codes:

'***' 0.001

'**' 0.01

'*' 0.05.

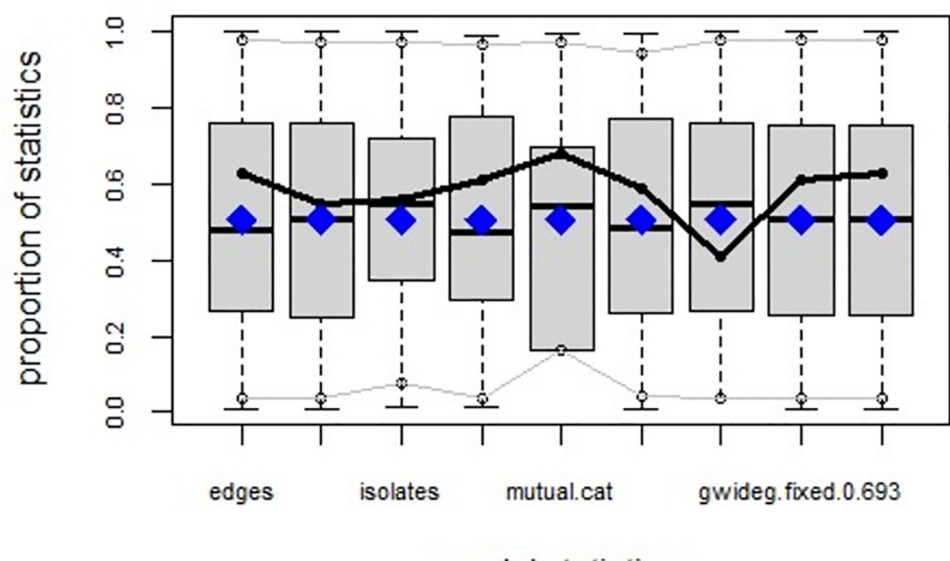

**Fig 7. Multiplexity in the alpha community—goodness of fit [24] model statistics.**

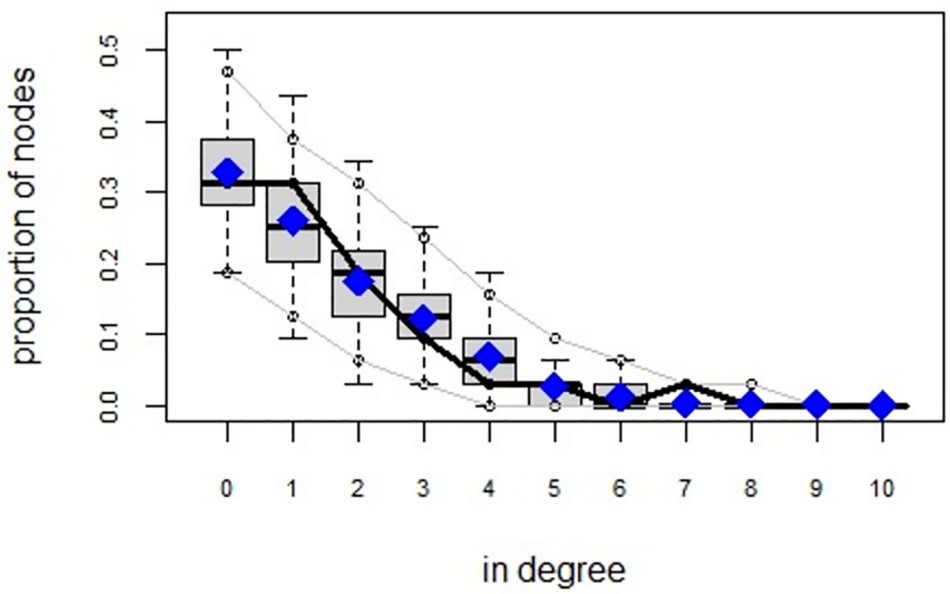

**Fig 8. Multiplexity in the alpha community—goodness of fit [24] in degree.**

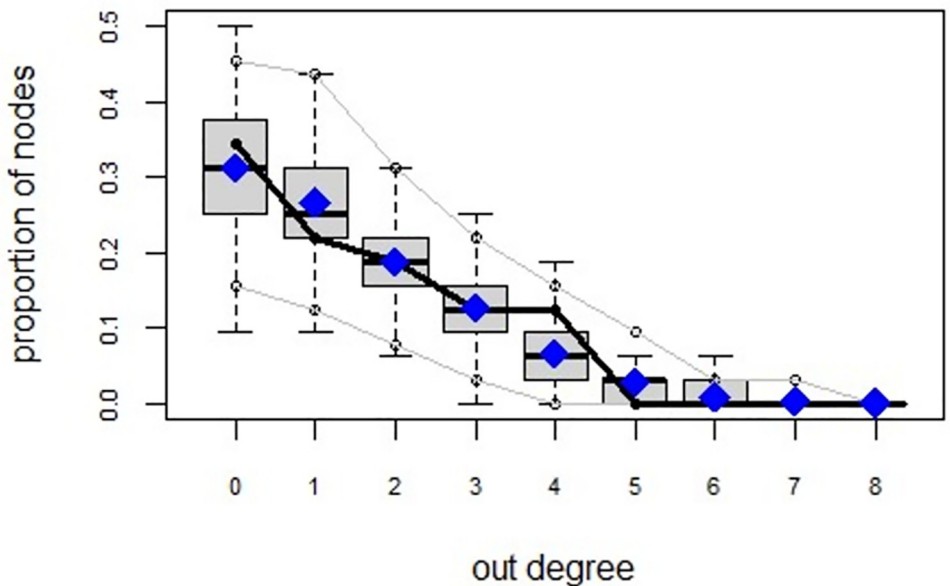

**Fig 9. Multiplexity in the alpha community—goodness of fit [24] out degree.**

The isolate effect is present because some nodes do not have multiplex links in the resultant digraph. The nodal degree -in/out- has been fixed as a consequence of the restriction in the sociometric generator. A lambda parameter (0.693) has been introduced for controlling possible effects of extreme nodal degree distribution We must remember that we have only

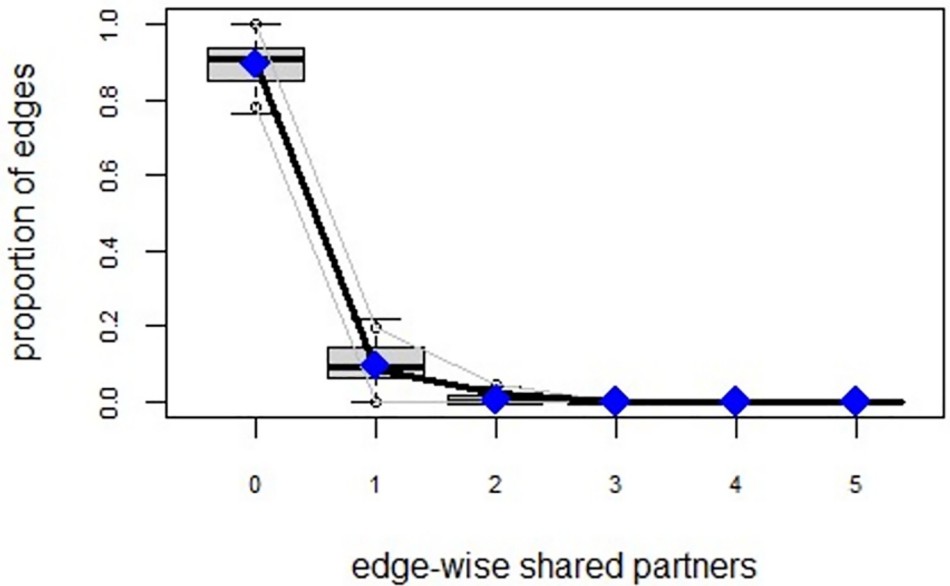

**Fig 10. Multiplexity in the alpha community—goodness of fit [24] edge-wise shared partners.**

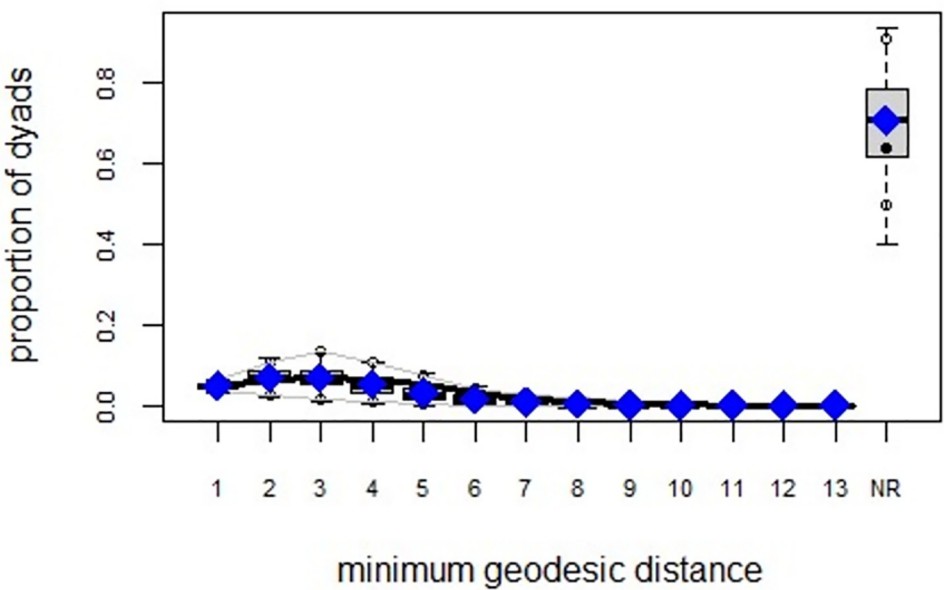

**Fig 11. Multiplexity in the alpha community—goodness of fit [24] minimum geodesic distance.**

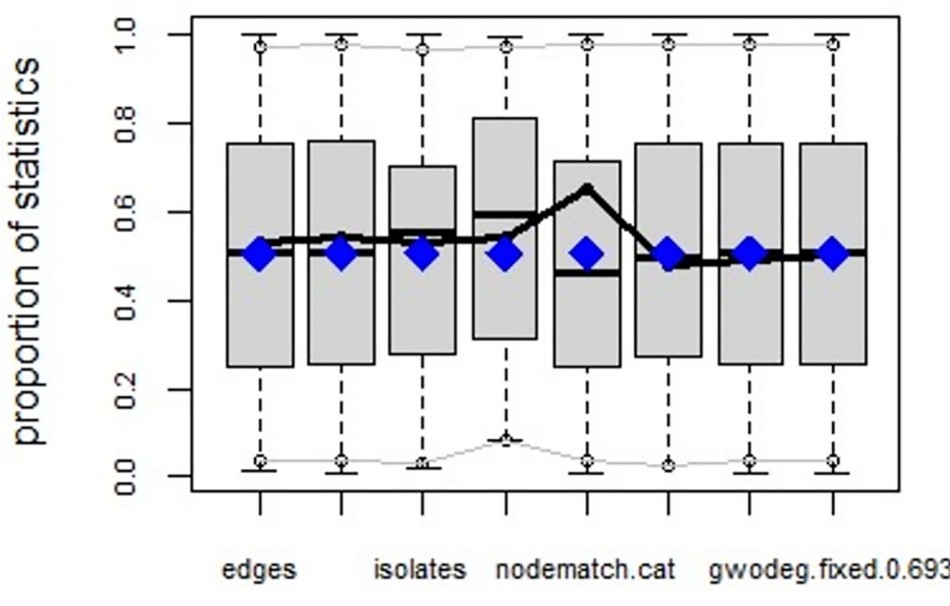

**Fig 12. Multiplexity in the beta community–goodness of fit [24] model statistics.**

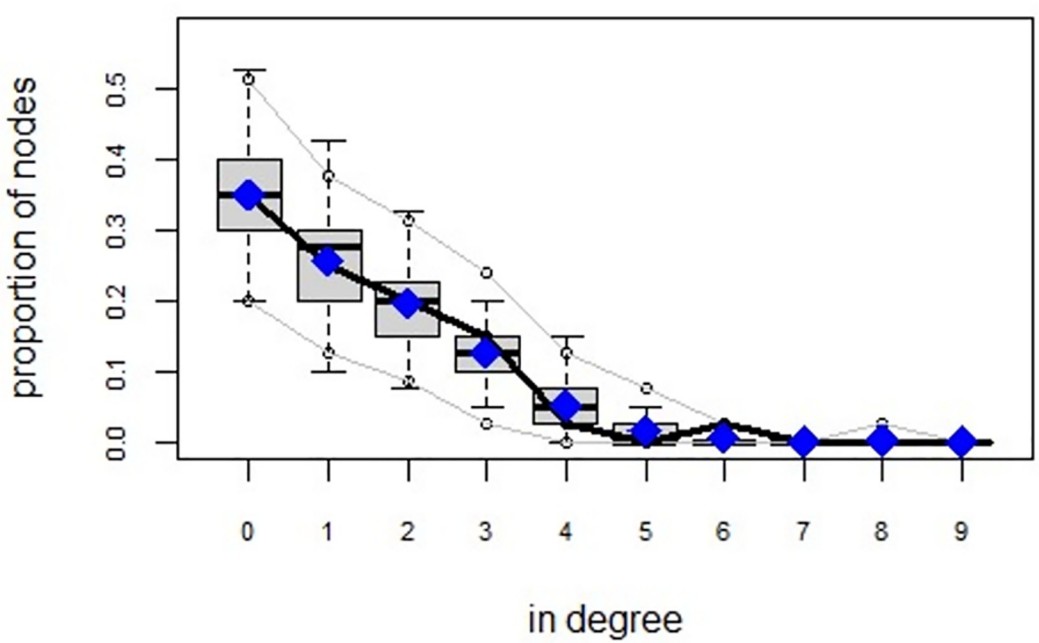

**Fig 13. Multiplexity in the beta community—goodness of fit [24] in degree.**

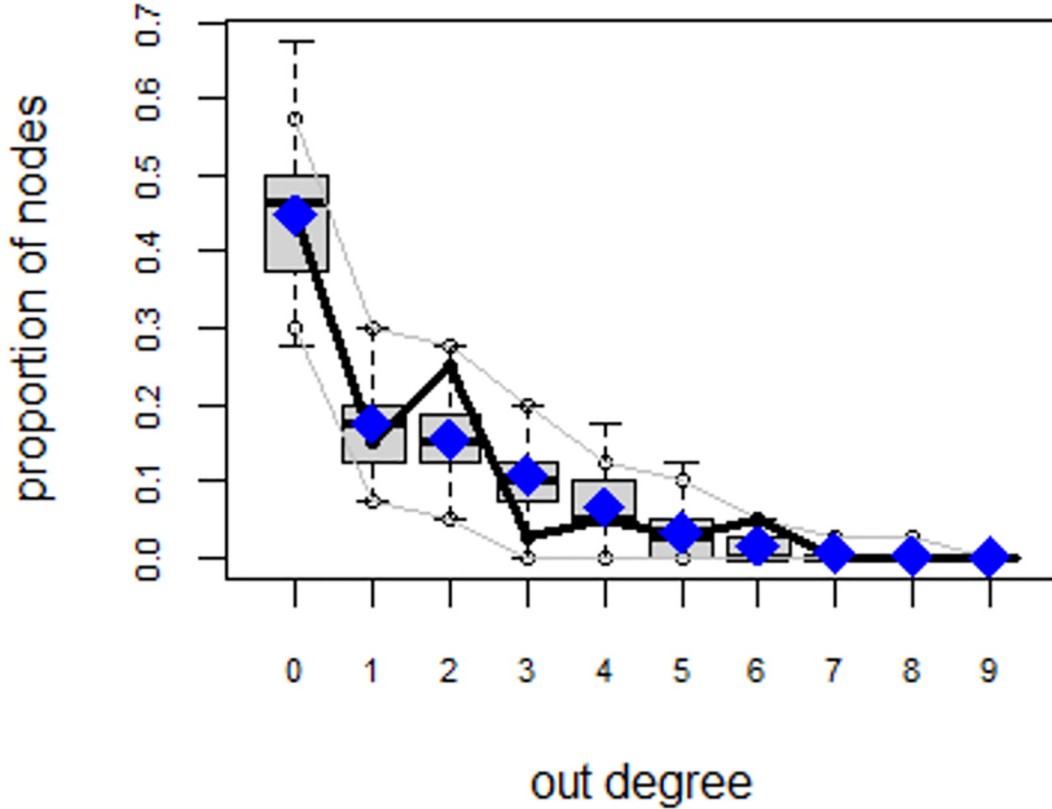

**Fig 14. Multiplexity in the beta community—goodness of fit [24] out degree.**

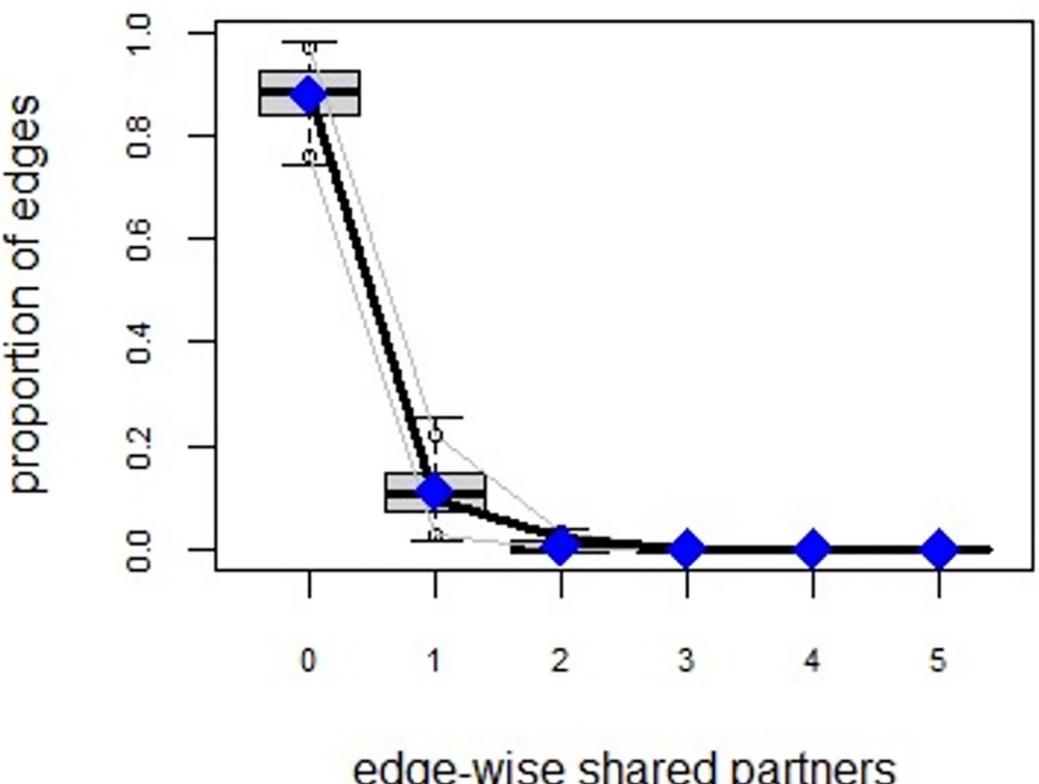

**Fig 15. Multiplexity in the beta community—goodness of fit [24] edge-wise shared partners.**

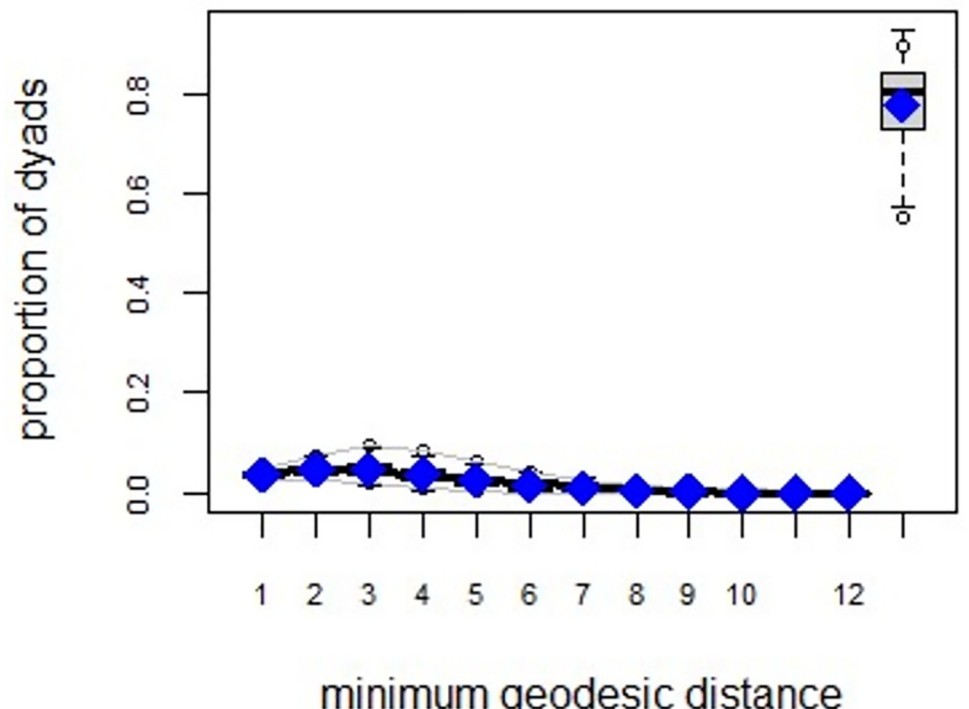

**Fig 16. Multiplexity in the beta community—goodness of fit [24] minimum geodesic distance.**

considered strict multiplexity—that is, the links that encompass the three processes simultaneously. In a statistical way, the results are:

Based on parameters and statistical significance, we can say that in both communities the strength of ties has a positive effect on multiplexity. In other words, the stronger the strength of interactions, the greater the probability of observing a multiplex tie. From the point of view of Granovetter's definition of strength [1] and Greenbaum's results [5], this result is expected. Nevertheless, there is something new in these communities: multiplexity is not simply the overlap of social circles. Differently from the standard concept of multiplexity, the conjunction of three different processes (status, exchange of information and collaboration) occurs in a segmented way, in clearly differentiated, that is specialized, realms (education, religion, politics, security, health, business, local organization, sports, and leisure), where the leaders exercise their authority and participate in mediation processes.

## Discussion and conclusion

Our findings reveal some facts about how social processes sustain cohesion in the high-order structure of elite connections. These results suggest a segmented multiplexity in specialized domains of performance which is fueled by both a high frequency of interactions and reciprocal relationships.

If we assume that elite connections make up of a high order structure, our findings, in contrast to well- established hypotheses, reveal a segmented social order in which multiplexity does not mean the overlapping of social circles. On the contrary, different social exchanges are restricted to well-differentiated domains. This is different to Greenbaum's idea of cohesion as a consequence of overlapping different social circles. At the same time, two elements appear that are not present in Granovetter's hypothesis [1]. On the one hand, a shared domain is a social circle patterned by common interests, understandings, and values that enhance strong ties. On the other hand, and despite the predictive effect of strong ties, the elite structure is not a fragmented network with several components. On the contrary, it is a unique connected component.

How could we explain these divergences? New research should operationalize community cohesion not simply as a formal property of interaction structures but as a social discipline [19]. In other words, it should point to the correspondence between the axiological universe, which gives order and meaning to social exchanges, and the structural forms that are highlighted by topological techniques. In the same way as White [19], we can say, in a preliminary way, that the multiplex structure, of a segmented type, hints at the emergence, among leaderships, of a social discipline of council characterized by a mediation process. This process is guided by prestige and is focused on the control of resources vital to the various domains of interaction. In fact, many leaders mediate in the search for financial and technical resources that satisfy the needs of the local population, appearing as natural providers of community life. However, delving into an analysis of social discipline requires other models and new empirical data.

## Supporting information

**S1 Dataset.**
(RAR)

**S1 File.**
(DOCX)

**S1 Graphics.**
(DOCX)

**S1 Table.**
(TXT)

## Author Contributions

**Conceptualization:** Silvio Salej Higgins.

**Formal analysis:** Neylson Crepalde.

**Methodology:** Silvio Salej Higgins.

**Validation:** Silvio Salej Higgins.

**Writing – original draft:** Silvio Salej Higgins.

**Writing – review & editing:** Ivan L. Fernandes.

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
