## [Decision Letter · Decision Letter 0]

25 Jan 2021

PONE-D-20-24819

Is made the social cohesion by weak ties or by multiplex ties? Rival hypotheses about leader networks in urban community settings

PLOS ONE

Dear Dr. Higgins,

Thank you for submitting your manuscript to PLOS ONE. After careful consideration, we feel that it has merit but does not fully meet PLOS ONE’s publication criteria as it currently stands. Therefore, we invite you to submit a revised version of the manuscript that addresses the points raised during the review process.

As you will see, both reviewers agree that there is merit in this work but they have identified specific problems and raised certain concerns related to the context and interpretation of the model, especially related to the original work of Granovetter. Also, please make sure that you correct the title of the paper which is currently grammatically wrong.

We look forward to receiving your revised manuscript.

Kind regards,

Lazaros K. Gallos

Academic Editor

PLOS ONE

Journal Requirements:

3. We note you have included tables to which you do not refer in the text of your manuscript. Please ensure that you refer to Table 3, 4, 11, and 12 in your text; if accepted, production will need this reference to link the reader to the Table.

Reviewers' comments:

Reviewer's Responses to Questions

**Comments to the Author**

1. Is the manuscript technically sound, and do the data support the conclusions?

Reviewer #1: Partly

Reviewer #2: Partly

2. Has the statistical analysis been performed appropriately and rigorously? 

Reviewer #1: I Don't Know

Reviewer #2: Yes

3. Have the authors made all data underlying the findings in their manuscript fully available?

Reviewer #1: No

Reviewer #2: Yes

4. Is the manuscript presented in an intelligible fashion and written in standard English?

Reviewer #1: No

Reviewer #2: Yes

5. Review Comments to the Author

Reviewer #1: This paper aims to empirically evaluate whether cohesion is facilitated by weak ties for multiplex ties. This is an interesting question, and the authors examine it in an interesting context, however I have several concerns about the current version:

(1) Throughout, the writing requires proofreading to correct grammatical issues. For example, the current title is "Is made the social cohesion by weak ties or by multiplex ties?" but should be something like "Is social cohesion produced by weak ties or by multiplex ties?"

(2) The data sharing statement is insufficient. If the authors have permission to share the data, then the data should be placed in a publicly accessible archive. If the authors do not have permission to share the data, this should be explained, and the third-party owner of the data should be identified.

(3) In Granovetter's original work, there was ambiguity concerning the definition of tie strength. Although Granovetter references things like interaction frequency or intensity, ultimately he defines a tie's strength in terms of whether it is a bridge. That is, for Granovetter, a tie's strength is a structural property (what we might now call edge betweenness) and not a property of the edge itself. This ambiguity continues in this paper, where you measure the strength of ties using an interaction frequency scale. This could be a perfectly reasonable approach, but is not how Granovetter conceptualied tie strength. This should be clarified.

(4) You operationalize tie strength using interaction frequency. However, it could also be reasonable to operationalize tie strength using multipliexity: more multiplex ties are stronger. I suspect that your tie strength and multiplexity measures are highly correlated, which could cause problems for model estimation and interpretation. It would be helpful if you could report the correlation between (a) a dyad's interaction frequency and (b) their multiplexity.

(5) When collecting your data, respondents were limited to 10 alters for each question. This imposes restrictions on each ego network, and on the whole network. For example, it imposes a maximum possible degree for each node, and it imposes upper limits on each of the metrics reported in Table 3. Best practice in network data collection is to allow respondents to name as many alters as they want, and not to impose such restrictions. However, because these data were collected with restrictions, they must be explicitly specified and modeled in the ERGM.

(6) The Digraphs are fuzzy and illegible.

(7) I had trouble understanding your models. Much of this could be clarified if you provided replication code, but in the absence of replication code: (A) Because you have data on tie strength, why not retain this information and estimate a generalized ERGM, which can use valued network data? (B) If your hypothesis concerns comparing the effect of tie strength and multiplexity, why is it necessary to consider each of the three types of ties separately? These additional analyses seem unrelated to your hypotheses and make the paper unnecessarily long.

(8) In your analysis of multiplexity in the alpha community, you write "The frequency of interactions, taken by us as a proxy of strength, reveals a difference between weak ties and strong ties." However, you do not explicitly test whether the effect of weak ties differs from the effect of strong ties. And, the observed difference between the two (92% vs. 87%) is negligible. Therefore, I do not think you can conclude that there is a difference between weak and strong ties. Although the observed difference is larger in the beta community, because there is no explicit test, you also cannot draw this conclusion there.

Reviewer #2: Is social cohesion made by weak ties or multiplex ties? Rival hypotheses regarding leader networks in urban community settings

This article addresses the sociological question of the factors underpinning community cohesion by analyzing original data from two areas of Brazil, with state-of-the-art ERGM models. The writing style is engaging, the data are interesting, the structure is clear, and the authors use the models appropriately.

My main concern is the conceptualization of the problem. The points made by Granovetter and his cited critics are sometimes presented in a slightly different light relative to the original version, and this obfuscates the meaning and scope of the analysis. First, it should be made clear that Granovetter and his critics are all talking about large communities, where members are so numerous that there cannot possibly be strong ties linking all of them. Otherwise, the question in the title would simply not make any sense: it would be obvious that strong ties ensure cohesion. Second, multiplex ties are just strong ties – whereby multiplicity of tie properties (in Greenbaum’s 1982 paper, neighborhood acquaintance and another such as kinship or shared occupation) replaces Granovetter’s measure of strength based on frequency, duration etc. So, in what respect is the opposition weak/multiplex ties new? Isn’t it just one of the many ways to assert “the strength of strong ties”, as many authors have already done since 1973? And, to go back to the first issue, aren’t multiplex ties more likely to occur in smaller communities – that is, settings different from Granovetter’s?

One way for the authors of this article to get around this problem could be to distinguish more explicitly between the relational dimension of social ties, however measured, and physical proximity – which would make sense in urban settings. Granovetter, Gans and Greenbaum all evoke a spatial dimension and suggest it is important, but they do not model it explicitly, so it remains fuzzy. There is a recent literature that develops on this point (eg Polge & Torre 2018, Torre et al 2019, Torre and Rallet 2005). I am not sure the authors have enough data to explore this aspect, but it would definitely be worthwhile – and in passing, it would help give a more precise definition of an “urban community”.

My other concern is that there is a slight mismatch between the stated problem (ties, or absence thereof, between grassroots and the elite – which was a major point in Gans’s work, but not in Granovetter’s) and what the authors’ data are about (ties between elite members). I would invite the authors to state much more precisely how their empirical analysis operationalizes their questions. In my view, what the authors study is the relationship between frequency of interaction (the measure they take for tie strength) and shared domains of activity and performance (which is at the basis of multiplexity). But, isn’t that just a way to say that both are measures of tie strength? That they are correlated, is not so surprising: if you interact with other people in multiple social circles, then inevitably you interact with them more often, and social cohesion builds on that. The authors will recognize that I have in mind Alba and Kadushin 1976.

In the data, frequency of contact has been measured for one set of ties only, but it seems interpreted as strength of ties overall. Perhaps results should be interpreted more cautiously (for example, someone named for collaboration but not for status may be a very frequent contact).

Some details:

The is a typo in the metadata (the title reads “Is made social cohesion…”).

PP. 9 and 13: Granovetter did write “In the absence of actual network data, all this is speculation” but he was referring specifically to the study of Gans, who hadn’t collected such data. He wasn’t suggesting that it is always challenging to sample a structured set of interactions. By the way, I am not sure what the authors mean when they refer to the problem of non-independence of observations in network data, p. 13. The problem is not to eliminate dependence, but to control or (better) to model it, as ERGM do.

P. 14, especially Table 1: what is meant precisely by “global” and “local”?

P. 15 The authors mention data on negative ties (conflict etc.) and bridging social capital without giving any details. But because they do not use these data, I suggest removing the mention at all, and focusing only on the data used in the analysis.

P. 19 The two groups of elite members are pretty small (32 and 40 people respectively). But how large are the communities of which they are the elites? This is important to understand the extent to which Granovetter’s framework applies (my comment above).

P. 24, What is the difference between the left and right columns of Table 5? The effects are the same, but parameter values, standard errors and t-ratios are not.

Why are the effects of the models for Beta different from those of Alpha?

What is the significance level chosen – 5%?

There is a typo in the Reciprocity effect of the left column: the t-ratio should not be negative.

PP. 24-25, The comments do not match the figures in the table. For example, the Strong Ties coefficient of Model B for community Alpha is 3.1819, but it is said that “strong ties […] increase the probability of forming multiplex links by 92%”: why? If any transformations are required to make the table values interpretable, they should be stated clearly.

References

Alba, R. D., & Kadushin, C. (1976). The intersection of social circles: A new measure of social proximity in networks. Sociological Methods & Research, 5(1), 77–102.

Polge, E. & Torre, A. (2018). Territorial governance and proximity dynamics. the case of two public policy arrangements in the Brazilian amazon. Papers in Regional Science, 97(4), 909–929.

Torre, A., Polge, E., & Wallet, F. (2019). Proximities and the role of relational networks in innovation: The case of the dairy industry in two villages of the “green municipality” of paragominas in the eastern amazon. Regional Science Policy & Practice, 11(2), 279–294.

Torre, A. & Rallet, A. (2005). Proximity and localization. Regional Studies, 39(1), 47–59.

6. PLOS authors have the option to publish the peer review history of their article (what does this mean?). If published, this will include your full peer review and any attached files.

Reviewer #1: **Yes: **Zachary P. Neal

Reviewer #2: No

---

## [Author Response · Author response to Decision Letter 0]

22 Jun 2021

All the comments have been responded in the attached rebuttal letter.

---

## [Decision Letter · Decision Letter 1]

28 Jul 2021

PONE-D-20-24819R1

Is social cohesion produced by weak ties or by multiplex ties?

Rival hypotheses regarding leader networks in urban community settings

PLOS ONE

Dear Dr. Higgins,

Thank you for submitting your manuscript to PLOS ONE. After careful consideration, we feel that it has merit but does not fully meet PLOS ONE’s publication criteria as it currently stands. Therefore, we invite you to submit a revised version of the manuscript that addresses the points raised during the review process.

As you will notice, we did not manage to secure the report of a second reviewer. However, the report that we received is already quite thorough and provides specific advice on how to improve your manuscript. In the interest of speeding up the process, I have decided to proceed with this one report and give you the chance to respond to those comments. In the next round of submission, I will most likely consult an additional reviewer.

We look forward to receiving your revised manuscript.

Kind regards,

Lazaros K. Gallos

Academic Editor

PLOS ONE

Reviewers' comments:

Reviewer's Responses to Questions

**Comments to the Author**

1. If the authors have adequately addressed your comments raised in a previous round of review and you feel that this manuscript is now acceptable for publication, you may indicate that here to bypass the “Comments to the Author” section, enter your conflict of interest statement in the “Confidential to Editor” section, and submit your "Accept" recommendation.

Reviewer #1: (No Response)

2. Is the manuscript technically sound, and do the data support the conclusions?

Reviewer #1: Partly

3. Has the statistical analysis been performed appropriately and rigorously? 

Reviewer #1: Yes

4. Have the authors made all data underlying the findings in their manuscript fully available?

Reviewer #1: Yes

5. Is the manuscript presented in an intelligible fashion and written in standard English?

Reviewer #1: Yes

6. Review Comments to the Author

Reviewer #1: The authors have done a good job addressing most of my original comments, however one of my more overarching original concerns remains. In my first review, I commented that multiplexity and frequency could both be viewed as measures of tie "strength." This is a point raised by Reviewer #2 also, and indeed Granovetter (1973) himself wrote that "most multiplex ties [are] strong" (p. 1361). Several features of the current paper confirm this point. For example, on page 5 you summarize Greenbaum's finding by mentioning "strong multiplex ties" and on page 17 you find that frequency and multiplexity are highly correlated at r > 0.4 (despite your claim to the contrary, I view this as a large correlation). For these reasons, I continue to believe that multiplexity and frequency are both measuring different aspects of a common construct called "strength". This raises at least two issues:

First, because others have previously used both frequency and multipliexity as indicators of strength, it is confusing that you treat frequency as an indicator of strength, but not multiplexity. It would be clearer if you identified both as indicators of strength, and throughout the paper simply examined the relationship between frequency and multiplexity.

Second, your finding that "the strength of ties [i.e. frequency] has a positive effect on multiplexity" seems obvious given prior work that has treated frequency and multiplexity as both indicators of strength, and given your own observed correlation between the two. You go on to claim that this finding is not trivial and that "there is something subtle here." Perhaps, but it is so subtle that I was unable to understand why this finding is not trivial. If the core finding of your analysis is that the association between frequency and multiplexity is *not* trivial, the reasons why it is not trivial need to be clearer.

As a more minor, but still important, point - Your analyses hinges upon results from two ERGM, however I did not see any evidence of these models' goodness-of-fit. This should be included to confirm that the model estimates are reasonable.

7. PLOS authors have the option to publish the peer review history of their article (what does this mean?). If published, this will include your full peer review and any attached files.

Reviewer #1: **Yes: **Zachary Neal

---

## [Author Response · Author response to Decision Letter 1]

27 Aug 2021

I have attached a specif letter with all details.

---

## [Decision Letter · Decision Letter 2]

7 Sep 2021

Is social cohesion produced by weak ties or by multiplex ties?

Rival hypotheses regarding leader networks in urban community settings

PONE-D-20-24819R2

Dear Dr. Higgins,

We’re pleased to inform you that your manuscript has been judged scientifically suitable for publication and will be formally accepted for publication once it meets all outstanding technical requirements.

Kind regards,

Lazaros K. Gallos

Academic Editor

PLOS ONE

Additional Editor Comments (optional):

Reviewers' comments:

Reviewer's Responses to Questions

**Comments to the Author**

1. If the authors have adequately addressed your comments raised in a previous round of review and you feel that this manuscript is now acceptable for publication, you may indicate that here to bypass the “Comments to the Author” section, enter your conflict of interest statement in the “Confidential to Editor” section, and submit your "Accept" recommendation.

Reviewer #1: All comments have been addressed

2. Is the manuscript technically sound, and do the data support the conclusions?

Reviewer #1: (No Response)

3. Has the statistical analysis been performed appropriately and rigorously? 

Reviewer #1: (No Response)

4. Have the authors made all data underlying the findings in their manuscript fully available?

Reviewer #1: (No Response)

5. Is the manuscript presented in an intelligible fashion and written in standard English?

Reviewer #1: (No Response)

6. Review Comments to the Author

Reviewer #1: (No Response)

7. PLOS authors have the option to publish the peer review history of their article (what does this mean?). If published, this will include your full peer review and any attached files.

Reviewer #1: **Yes: **Zachary P. Neal

---

## [Editor Report · Acceptance letter]

10 Sep 2021

PONE-D-20-24819R2 

Is social cohesion produced by weak ties or by multiplex ties? Rival hypotheses regarding leader networks in urban community settings  

Dear Dr. Higgins:

I'm pleased to inform you that your manuscript has been deemed suitable for publication in PLOS ONE. Congratulations! Your manuscript is now with our production department. 

Kind regards, 

on behalf of

Dr. Lazaros K. Gallos 

Academic Editor

PLOS ONE